# Oral Supplementation of L-Carnosine Attenuates Acute-Stress-Induced Corticosterone Release and Mitigates Anxiety in CD157 Knockout Mice

**DOI:** 10.3390/nu16172821

**Published:** 2024-08-23

**Authors:** Takahiro Tsuji, Kazumi Furuhara, Erchu Guo, Yijing Wu, Jing Zhong, Haruhiro Higashida, Yasuhiko Yamamoto, Chiharu Tsuji

**Affiliations:** 1Research Center for Child Mental Development, Kanazawa University, Kanazawa 920-8640, Japan; furururukz.999@gmail.com (K.F.); guoerchu123@gmail.com (E.G.); yaya763687757@163.com (Y.W.); haruhiro@med.kanazawa-u.ac.jp (H.H.); 2Department of Ophthalmology, Faculty of Medical Sciences, University of Fukui, Fukui 910-1193, Japan; 3Life Science Innovation Center, University of Fukui, Fukui 910-1193, Japan; 4Departments of Biochemistry and Molecular Vascular Biology, Kanazawa University Graduate School of Medical Sciences, Kanazawa 920-8640, Japan; yasuyama@med.kanazawa-u.ac.jp; 5School of Pharmacy, Guangxi University of Chinese Medicine, Nanning 530011, China; 6Physiological Department, Guangxi University of Chinese Medicine, Nanning 530011, China; zhongjing1212@163.com; 7Department of Socioneurosciences, United Graduate School of Child Development, Osaka University, Kanazawa University, Hamamatsu University School of Medicine, Chiba University and University of Fukui, Kanazawa Campus, Kanazawa 920-8640, Japan

**Keywords:** L-carnosine, CD157, autism spectrum disorder, corticosterone, anxiety, stress

## Abstract

Corticosterone, an end product of the hypothalamic–pituitary–adrenal (HPA) axis, is a crucial stress hormone. A dysregulated HPA axis and corticosterone release play pivotal roles in the onset and persistence of symptoms of stress-related psychiatric disorders, such as anxiety. The intake of nutrients, probiotics, and prebiotic supplements decreases blood corticosterone levels. The dipeptide L-carnosine is composed of beta-alanine and L-histidine and is commercially available as a nutritional supplement for recovery from fatigue. L-carnosine is involved in stress-induced corticosterone responses and anxiety behaviors in rodents. Here, we assessed the effect of L-carnosine in CD157 knockout (KO) mice, a murine model of autism spectrum disorder (ASD). The uptake of L-carnosine suppressed the increase in plasma corticosterone levels in response to acute stress and attenuated anxiety-like behaviors in CD157 KO mice. These results suggest that L-carnosine supplementation may relieve anxiety by suppressing excessive stress responses in individuals with ASD.

## 1. Introduction

Organisms adapt to stress by stimulating the hypothalamic-pituitary-adrenal (HPA) axis to cope with environmental changes. The adrenal glands secrete corticosterone, the primary glucocorticoid in humans, which increases pulse and blood pressure, raises blood sugar levels, and suppresses excessive immune responses. However, excessive corticosterone levels weaken the prefrontal cortex and cause neuronal death in the hippocampus, leading to anxiety, depression, and other neurological and psychiatric disorders. Therefore, attempts have been made to reduce surplus corticosterone levels [1].

Several nutrients, probiotics, and prebiotics are potential therapeutic agents for mitigating excessive stress responses [2]. L-carnosine is an imidazole dipeptide composed of beta-alanine and L-histidine, and its concentration is very high in the skeletal muscle and brain of mammals. It can cross the blood–brain barrier or can be synthesized from the beta-alanine and histidine in the brain. In practice, L-carnosine has been marketed as a nutritional supplement for its anti-aging and fatigue-relieving properties and ameliorating effects on lifestyle-related diseases (e.g., diabetes, hypertension, and atherosclerosis) through its antioxidant and pH-buffering properties [3,4]. In the brain, it has beneficial effects on various neuropsychiatric disorders such as ischemic stroke, cognitive impairment [5,6], autism spectrum syndrome [7], schizophrenia [8,9], Alzheimer’s disease and dementia [10,11], attention deficit hyperactivity disorder [12], and Gulf War syndrome [13]. L-carnosine is involved in stress-induced corticosterone responses and anxiety behaviors in rodents. L-carnosine-administered mice subjected to restraint stress showed suppressed elevation of plasma corticosterone levels compared to that in the control Kunming mice [14,15].

Bone marrow stromal cell antigen (BST-1), also known as CD157, was first cloned as a glycosyl phosphatidylinositol-anchored protein involved in the growth of pre-B cells [16]. CD157/BST-1 is a glycosyl phosphatidylinositol-anchored membrane protein that functions as an ADP ribosyl cyclase, and the loss of CD157 expression in mice results in anxiety-like behaviors and social behavioral deficits. It is a paralog of CD38 that catalyzes cyclic ADP-ribose to regulate intracellular Ca^2+^ [17]. CD157/BST-1 is constitutively expressed in the myeloid cells of peripheral blood mononuclear cells and regulates the humoral immune response [18]. Moreover, CD157/BST-1 is associated with neuropsychiatric disorders, such as Parkinson’s disease, autism spectrum disorder (ASD), rapid eye movement sleep behavior disorder, major depressive disorder, restless leg syndrome/Willis–Ekbom disease, and Alzheimer’s disease [19]. Although the physiological role of CD157 in the brain remains largely unexplored, an association between CD157/BST1 and ASD has been reported [20,21]. Homozygous CD157 knockout (CD157 KO) mice display social behavioral impairments and anxiety-related and depression-like behaviors, which can be restored by treatment with antidepressants or oxytocin [22,23,24]. These findings suggest that CD157 KO mice may be useful as a model of ASD with regard to modeling the behaviors associated with ASD symptoms. We previously observed that the chronic administration of L-carnosine ameliorates social behavioral deficits, which is a core symptom of ASD, in CD157 KO mice [25]. However, the effects of L-carnosine on other comorbid symptoms of ASD, such as anxiety-related behaviors and altered stress responses, have not yet been investigated in CD157 KO mice. In this study, we examined the effect of chronically administered L-carnosine on the corticosterone response induced by acute stress and anxiety-like behavior in CD157 KO mice.

## 2. Materials and Methods

### 2.1. Animals

C57BL6/N wild-type (WT) mice were obtained from Japan SLC Inc. (Hamamatsu, Japan) via Sankyo Laboratory Service Corporation (Toyama, Japan). CD157 KO mice were developed as previously described [18]. Homozygous CD157 KO mice were used in this study. WT and CD157 KO mice were housed at the Institute for Experimental Animals, Advanced Science Research Center, Kanazawa University, under standard conditions (22 °C; 12-h light/dark cycle, lights on at 8:45 a.m.) in standard mouse cages (300 × 160 × 110 mm) with sawdust bedding and access to food and water ad libitum. Mice weaned at 21–28 days of age were housed in same-sex groups of three to five animals until 11 weeks of age. Male mice were single-housed for 14 days before acute stress and behavioral tests were conducted. Carnosine-treated mice were maintained on a steady dose of L-carnosine (Phytopharma Co., Ltd., Yokohama, Japan) diluted in drinking water (0.09 g/100 mL) from weaning until the behavioral test. Water intake and body weight were measured during the experiment. This study was conducted in accordance with the Fundamental Guidelines for Proper Conduct of Animal Experiment and Related Activities in Academic Research Institutions under the jurisdiction of the Ministry of Education, Culture, Sports, Science and Technology of Japan. The protocol was approved by the Committee on Animal Experimentation of Kanazawa University (AP-143261, 11 April 2022). We divided the mice into three groups, wild-type (WT) as control, CD157 KO mice with or without L-carnosine administration (KO-Car or KO-Water). The number of mice used in this study is described in the text and figures.

### 2.2. Acute Stress

Forced swimming was performed as previously described [22]. Briefly, mice were placed individually in a cylinder (height 25 cm, diameter 15 cm) filled up to a 10 cm depth with water (25 ± 1 °C) for 6 min. A restraint stress study was performed by placing the mice in a 50 mL polypropylene conical tube (Eppendorf, Hamburg, Germany) with air holes for 10 min.

### 2.3. Plasma Sampling and Enzymatic Detection of Corticosterone 

Male mice were anesthetized immediately after acute stress by an intraperitoneal injection of pentobarbital (35 mg/kg). Blood samples (0.8–1 mL) were collected by cardiac puncture, and 8–10 μL of 0.1 g/mL ethylenediaminetetraacetic acid was added. The samples were centrifuged at 1600× *g* for 15 min at 4 °C. Plasma samples (200–400 μL/mouse) were collected and stored at −80 °C until use.

### 2.4. Enzyme Immunoassay of Corticosterone

Immunoreactivity of plasma corticosterone was analyzed using a corticosterone ELISA kit (Cayman Chemical, Ann Arbor, MI USA), following the manufacturer’s instructions. The plasma samples (5 μL) were thawed and diluted to 1:80 in assay buffer. Fifty microliters of the sample was used for the assay. Blood samples were assayed without protein extraction, as previously described [24]. The assay had two linear ranges covering a concentration range of 30–1000 pg/mL. The inter- and intra-assay coefficients of variation were <5%.

### 2.5. Elevated Plus Maze

The mice in the home cage were placed in the experiment room for at least one hour for habituation. Duration of elevated plus maze was five minutes. Behavior was measured using digital video system and ANY-maze software v. 6.35 (Sloelting Co., Wood Dale, IL, USA).

### 2.6. Statistical Analysis

Statistical analysis was performed using Prism v.8 (GraphPad Software Inc., San Diego, CA, USA). The data are presented as mean ± standard deviation in the text. The induction of corticosterone in the plasma after the stress tests was compared with the baseline plasma corticosterone levels in each group using an unpaired *t*-test. One-way analysis of variance (ANOVA) was used to assess the differences in plasma corticosterone levels among groups. Subsequently, Bonferroni’s post hoc multiple comparison test was performed. Since we did not observe any effect of L-carnosine on plasma corticosterone levels in WT mice, no further behavioral examination of WT mice was conducted. For the behavioral examination, an unpaired *t*-test was conducted to compare each index between the KO-Water, KO-Car, and WT-Water groups.

## 3. Results

### L-Carnosine Mitigated Forced Swimming or Restraint-Stress-Induced Elevation in Plasma Corticosterone Levels

Acute stress increases plasma corticosterone levels. The effects of L-carnosine intake on the plasma corticosterone levels after forced swimming were examined in the WT (WT-W) and CD157 KO mice with (KO-C) and without (KO-W) chronic L-carnosine intake. While the basal corticosterone levels did not change between the groups (Figure 1a, 5.3 ± 3.4 ng/μL (*n* = 8), 8.4 ± 4.8 ng/μL (*n* = 10), and 10.1 ± 7.4 ng/μL (*n* = 6) for WT-W, KO-W, and KO-C, respectively), forced swimming stress increased the plasma corticosterone levels in all groups (control vs. forced swimming, *p* < 0.0001 in all four groups, unpaired *t*-test, Figure 1b, 57.6 ± 11.1 ng/μL (*n* = 8), 74.8 ± 12.2 ng/μL (*n* = 11), and 57.8 ± 16.5 ng/μL (*n* = 9) for WT-W, KO-W, and KO-C, respectively). The corticosterone levels in the KO-W mice were higher than those in the WT mice, whereas the corticosterone levels in the KO-C mice were similar to those in WT-W mice. One-way ANOVA revealed a significant difference in corticosterone levels between the groups (F [2,25] = 5.420, *p* = 0.011), and post hoc analysis with Bonferroni’s multiple comparisons test revealed significant differences in the corticosterone levels between the WT-W and KO-W groups and between the KO-W and KO-C groups (WT-W vs. KO-W, *p* = 0.032; KO-W vs. KO-C, *p* = 0.028).

Next, we examined the effect of L-carnosine on restraint stress. A single bout of restraint stress caused an increase in plasma corticosterone levels from the baseline values (Figure 1a) in all groups (*p* < 0.0001 in each group, unpaired *t*-test, Figure 1c, 49.2 ± 11.8 ng/μL (*n* = 19), 60.0 ± 18.7 ng/μL (*n* = 26), and 42.9 ± 12.1 ng/μL (*n* = 19) for WT-W, KO-W, and KO-C, respectively). The corticosterone levels in the KO-W mice were higher than those in the WT-W mice, whereas the corticosterone levels in the KO-C mice were similar to those in the WT-W mice. One-way ANOVA revealed a significant difference in the corticosterone levels between groups (F [2,61] = 7.375, *p* = 0.001), and post hoc analysis with Bonferroni’s multiple comparisons test revealed a tendency to a significant difference between WT-W and KO-W mice (*p* = 0.065) and a highly significant difference between KO-W and KO-C mice (*p* = 0.001). These results showed that the CD157 KO mice were more responsive than the WT mice to acute physical stress and that the oral supplementation with L-carnosine of CD157 KO mice mitigated the acute-stress-induced increases in blood corticosterone levels.

We examined whether L-carnosine uptake ameliorated anxiety-like behavior in CD157 KO mice. Anxiety was assessed using the elevated plus maze assay. The number of entries in the open arm decreased in the KO-W mice compared to that in the WT-W mice (*p* = 0.067, unpaired *t*-test) but increased in the KO-C mice compared to that in the KO-W mice (*p* = 0.061, unpaired *t*-test, Figure 2a, 5.0 ± 5.1 (*n* = 15), 1.9 ± 2.4 ng/μL (*n* = 12), and 4.5 ± 3.8 (*n* = 13) for WT-W, KO-W, and KO-C, respectively). The time spent in the open arms by the KO-W mice was shorter than that of the WT-W and KO-C mice (*p* = 0.033 and 0.0073, unpaired *t*-test, respectively), and these were at a similar level between KO-C and WT-W mice (*p* = 0.85, unpaired *t*-test, Figure 2b, 56.2 ± 69.2 s (*n* = 15), 10.2 ± 14.5 s (*n* = 12) and 39.6 ± 31.6 s (*n* = 13), WT-W, KO-W and KO-C, respectively). Although one-way ANOVA revealed a significant difference between groups (F [2,37] = 3.242, *p* =0.050), post hoc analysis with Bonferroni’s multiple comparisons test showed a significant difference in the corresponding test values only between the WT-W and KO-W mice (*p* = 0.0468). On the other hand, the time spent in the closed arm was not different among any of the groups (*p* = 0.49, WT-W vs. KO-W, and *p* = 0.79, KO-W vs. KO-C, unpaired *t*-test, Figure 2b, 137.9 ± 97.3 s (*n* = 15), 163.3 ± 93.0 s (*n* = 12), and 154.6 ± 71.0 s (*n* = 13) for WT-W, KO-W, and KO-C, respectively). These results demonstrated that L-carnosine ameliorated anxiety-like behavior in ASD mice.

## 4. Discussion

Acute stress significantly increased corticosterone release in CD157 KO mice, whereas CD157 KO mice that were chronically administered L-carnosine showed corticosterone levels similar to those of WT mice. Furthermore, L-carnosine treatment reduced anxiety-like behaviors in CD157 KO mice.

We previously reported that L-carnosine uptake improves social recognition behavior deficits in CD157 KO mice, probably through the activation of oxytocin neurons in the hypothalamus and increased secretion of oxytocin [25]. Therefore, L-carnosine may reduce corticosterone secretion through the activation of the oxytocinergic pathway. Exposure to various physiological and psychological stressors (immobilization, shaking, social defeat, forced swimming, or intracerebroventricular infusion of corticotropin-releasing factor [CRF]) can activate oxytocin neurons and facilitate the release of oxytocin in rodents [26,27]. Oxytocin innervates CRF neurons in the paraventricular nucleus to inhibit their activation, thereby inhibiting CRF secretion [27]. Exogenous oxytocin reduces CRF secretion and mitigates physical and mental responses to acute stress [28,29,30]. Furthermore, oxytocin neurons modulate CRF neurons and project to the vagus nerve and solitary bundle nuclei, thereby stimulating the parasympathetic neurons and directly relieving stress [31]. Therefore, the administration of L-carnosine may relieve acute physical stress by decreasing corticosterone secretion through the oxytocinergic pathway.

Anxiety disorders occur frequently among individuals with ASD, with a meta-analysis estimating that approximately 40% of ASD youths having at least one comorbid DSM-IV anxiety disorder [32]. Children with ASD have higher anxiety levels than typically developing children, and anxiety levels increase with intelligence quotient and age [33]. According to another meta-analysis, in autistic adults, although the prevalence rate is inconsistent depending on study design, the estimated current prevalence rate of anxiety is high, up to 27% [34]. Numerous studies have investigated cortisol responsiveness under physiological and/or psychosocial stress. Studies examining cortisol diurnal rhythms and cycles (cortisol arousal response, diurnal decline, and variability) indicate that relatively low-functioning individuals with autism show values different from those of typically developing individuals but do not consistently show similar changes in high-functioning individuals with autism [35]. In physically stressful environments, such as blood drawing, and mock magnetic resonance imaging scan, individuals with ASD respond more excessively to stressors than typically developed children. Furthermore, regarding psychosocial stressors, high reactivity has been observed in the playground during interactions with unfamiliar peers or short separations from a guardian [35]. Therefore, the hypersecretion of cortisol can be predicted for some psychological or non-psychological stress in people with autism. Our study suggests that the chronic intake of L-carnosine may reduce the dysregulated responsiveness of the HPA axis and help dampen the stress response in individuals with ASD.

One limitation of our study is that we only explored the effect of oral supplementation of L-carnosine from weaning to adulthood. Adolescence may be a sensitive time window for improving neuronal circuitry deficits. The environmental enrichment experienced during adolescence in rodents could restore the behavioral and emotional deficits induced by aversive experiences during the early postnatal age. Therefore, further investigations are necessary regarding the duration and time window of L-carnosine supplementation for understanding the underlying role of L-carnosine in anxiolytic effects and stress responses. Another limitation is that we used the ELISA assay to measure corticosterone level. Value obtained from ELISA assay may have certain biases due to sample matrix effects and cross-reactivity with other hormones. However, a correlation has been shown between LC-MS/MS and ELISA, although it could be disturbed in patients taking a corticosteroid synthesis inhibitor such as metyrapone or some steroid compounds [36]. Therefore, we used the ELISA assay in this study, which is generally admitted as a practical method for quantification.

## 5. Conclusions

This is the first study to demonstrate the anxiolytic effect of L-carnosine and the suppression of stress-induced corticosterone secretion by L-carnosine in an ASD mouse model. We found that L-carnosine supplementation may relieve anxiety by suppressing stress-induced hyperresponsivity, which appears in a subgroup of individuals with ASD.

## Figures and Tables

**Figure 1 nutrients-16-02821-f001:**
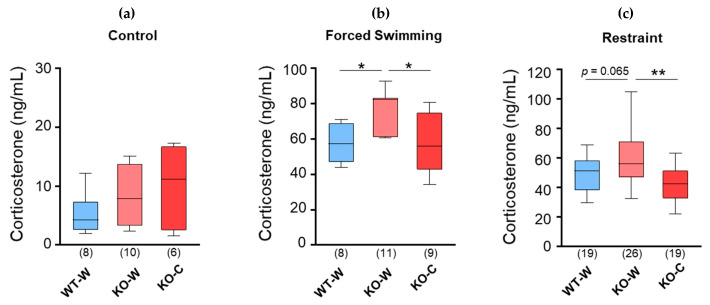
Plasma corticosterone level after acute stress in WT and CD157KO group with or without treatment of carnosine. (**a**) The plasma corticosterone without any stress as control. (**b**) The plasma corticosterone level after 6 min of forced swimming. (**c**) The plasma corticosterone levels after 6 min of restrained stress. Numbers of animals are shown in bars. Bars are represented in median and interquartile ranges in each group. * *p* ≤ 0.05, ** *p* ≤ 0.01.

**Figure 2 nutrients-16-02821-f002:**
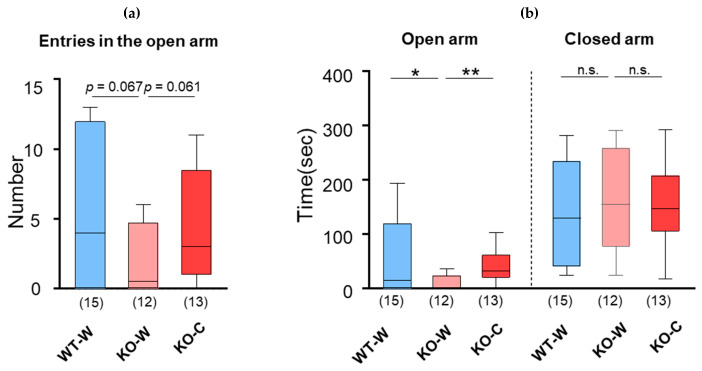
Elevated plus maze test. (**a**) Numbers of entries in the open arms. (**b**) Time in the open arm or the closed arm. Numbers of animals are shown in bars. Bars are represented in median and interquartile ranges in each group. * *p* ≤ 0.05. ** *p* ≤ 0.01. n.s.: not significant.

## Data Availability

The original contributions presented in the study are included in the article, further inquiries can be directed to the corresponding authors.

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
