# Peer review of "Oral Supplementation of L-Carnosine Attenuates Acute-Stress-Induced Corticosterone Release and Mitigates Anxiety in CD157 Knockout Mice"

_nutrients, 2024, doi:10.3390/nu16172821_

Round 1

Reviewer 1 Report

Comments and Suggestions for Authors

The manuscript submitted for review: "Oral supplementation of L-carnosine attenuates acute stress- induced corticosterone release and mitigates anxiety in CD157 knockout mice" is interesting and the conclusions raise some hopes. The authors mentioned in the discussion that L-carnosine changes the level of corticosterone secreted by oxytocin. It is a pity that they did not also determine the level of oxytocin. Why could not the level of oxytocin be determined?

Author Response

Comments : The manuscript submitted for review: "Oral supplementation of L-carnosine attenuates acute stress- induced corticosterone release and mitigates anxiety in CD157 knockout mice" is interesting and the conclusions raise some hopes. The authors mentioned in the discussion that L-carnosine changes the level of corticosterone secreted by oxytocin. It is a pity that they did not also determine the level of oxytocin. Why could not the level of oxytocin be determined?

Reply : 

We established the condition to detect oxytocin release in social scenes but not stressful scenes. Babrtedygitija et al. reported that not a single and acute restraint stress, but repeated restraint stress sufficient to release oxytocin in both blood andhypothalamus[1]. Further study remains necessary to examine the effect of L-carnosine on oxytocin release in stressful conditions enough to release oxytocin.

  1. Babygirija, R.; Bulbul, M.; Yoshimoto, S.; Ludwig, K.; Takahashi, T. Central and peripheral release of oxytocin following chronic homotypic stress in rats. Auton Neurosci 2012, 167, 56-60, doi:10.1016/j.autneu.2011.12.005.

Reviewer 2 Report

Comments and Suggestions for Authors

The manuscript nutrients-3138074 entitled Oral supplementation of L-carnosine attenuates acute stress-induced corticosterone release and mitigates anxiety in CD157 knockout mice by Takahiro Tsuji and coworkers, investigated the effect of L-carnosine in CD157 knockout (KO) mice, a murine model of autism spectrum disorder (ASD). The uptake of L-carnosine suppressed the increase in plasma corticosterone levels in response to acute stress and attenuated anxiety-like behaviors in CD157 KO mice. These results suggest that L-carnosine supplementation may relieve anxiety by suppressing excessive stress responses in individuals with ASD.

The scientific background is interesting and the introduction well focused on the topic.

The references are appropriate.

The narrative quality is good.

Material and methods are well described but

1. corticostrone is measured by ELISA and not LCMS/MS

2. results should be presented as mean ± standard deviation

3. Figures are informative but instead than bar plot, whiskers plot should be used.

The discussion is consistent with results.

Language is rather good.

Other points

Line 115: 2.4. Enzyme immunoassay of corticosterone. Immunoreactivity of plasma corticosterone was analyzed using a corticosterone EIA 116 kit (Enzo Life Sciences, NY, USA), following the manufacturer’s instructions.

The corticosterone quantification by immunoassay is prone to interferences. Isotope dilution mass spectrometry by LC-MS/MS is the reference method for corticosterone quantification.

This should be acknowledged in the discussion.

Line 129: The data are presented as mean ± standard error.

Results should be presented as mean ± standard deviation.

Standard error of the mean is used only in inferential statstic and in this case should not be used.

Line 133: We did not observe any effect of L-carnosine on blood corticosterone levels in WT mice; therefore, no further behavioral examination of WT mice was conducted.

It should be rewritten as:

Since we did not observe any effect of L-carnosine on blood corticosterone levels in WT mice, thus, no further behavioral examination of WT mice was conducted.

Also this sentence since to be in contrast with the results and the Authors should evaluate if save or remove.

Line 165 and line 183: the two graphics in the figures should be presented as bar and whiskers plot instead than a Bar plot.

Comments on the Quality of English Language

Language is rather good.

Author Response

Comments1: 

  1. corticostrone is measured by ELISA and not LCMS/MS
  2. Line 115: 2.4. Enzyme immunoassay of corticosterone. Immunoreactivity of plasma corticosterone was analyzed using a corticosterone EIA 116 kit (Enzo Life Sciences, NY, USA), following the manufacturer’s instructions.

    The corticosterone quantification by immunoassay is prone to interferences. Isotope dilution mass spectrometry by LC-MS/MS is the reference method for corticosterone quantification.

    This should be acknowledged in the discussion.

Response1: These two comments are similar. We responded to it below.

We added the sentence below in line 292 and a new reference, highlighted in yellow.

Another is methodology to measure corticosterone in serum. In this study, we used ELISA assay to measure corticosterone in serum. This method is generally admitted as a standard method for quantification. Although there is a high correlation between LC-MS/MS and ELISA, the latter has a certain bias due to sample matrix effect and cross-reactivity with other hormone [36]. Thus, LC-MS/MS is more reliable method and ELISA can be practically more preferable.

  1. Karachaliou, C.E.; Koukouvinos, G.; Goustouridis, D.; Raptis, I.; Kakabakos, S.; Petrou, P.; Livaniou, E. Cortisol Immunosensors: A Literature Review. Biosensors (Basel) 2023, 13, doi:10.3390/bios13020285.

Comments2: 

  1. results should be presented as mean ± standard deviation

Response2:

We wrote data as mean ± standard deviation in results section, highlighted in yellow.

Comments3: 

  1. Figures are informative but instead than bar plot, whiskers plot should be used.

Response3:

We re-produced figures to whiskers plot.

Comments4: 

Line 129: The data are presented as mean ± standard error.

Results should be presented as mean ± standard deviation.

Standard error of the mean is used only in inferential statstic and in this case should not be used.

Response4: We changed figures to whisker plot and wrote the values of mean ± standard deviation in Results section.

Comments5: 

Line 133: We did not observe any effect of L-carnosine on blood corticosterone levels in WT mice; therefore, no further behavioral examination of WT mice was conducted.

It should be rewritten as:

Since we did not observe any effect of L-carnosine on blood corticosterone levels in WT mice, thus, no further behavioral examination of WT mice was conducted.

Also this sentence since to be in contrast with the results and the Authors should evaluate if save or remove.

Response5: 

We corrected the sentences line187-188, highlighted in yellow.

Comments6: 

Line 165 and line 183: the two graphics in the figures should be presented as bar and whiskers plot instead than a Bar plot.

Response6: 

We changed the graphs to box and whiskers plots, reflecting median and interquartile ranges.
